# Designing light-element materials with large effective spin-orbit coupling

Jiayu Li[1], Qiushi Yao[1], Lin Wu[2,3], Zongxiang Hu[1], Boya Gao[1], Xiangang Wan [2,3✉] & Qihang Liu [1,4,5✉]

Spin-orbit coupling (SOC), which is the core of many condensed-matter phenomena such as nontrivial band gap and magnetocrystalline anisotropy, is generally considered appreciable only in heavy elements. This is detrimental to the synthesis and application of functional materials. Therefore, amplifying the SOC effect in light elements is crucial. Herein, focusing on $3d$ and $4d$ systems, we demonstrate that the interplay between crystal symmetry and electron correlation can significantly enhance the SOC effect in certain partially occupied orbital multiplets through the self-consistently reinforced orbital polarization as a pivot. Thereafter, we provide design principles and comprehensive databases, where we list all the Wyckoff positions and site symmetries in all two-dimensional (2D) and three-dimensional crystals that could have enhanced SOC effect. Additionally, we predict nine material candidates from our selected 2D material pool as high-temperature quantum anomalous Hall insulators with large nontrivial band gaps of hundreds of meV. Our study provides an efficient and straightforward way for predicting promising SOC-active materials, relieving the use of heavy elements for next-generation spin-orbitronic materials and devices.

[1] Shenzhen Institute for Quantum Science and Engineering (SIQSE) and Department of Physics, Southern University of Science and Technology, Shenzhen 518055, China. [2] National Laboratory of Solid State Microstructures and School of Physics, Nanjing University, Nanjing 210093, China. [3] Collaborative Innovation Center of Advanced Microstructures, Nanjing University, Nanjing 210093, China. [4] Shenzhen Key Laboratory of Advanced Quantum Functional Materials and Devices, Southern University of Science and Technology, Shenzhen 518055, China. [5] Guangdong Provincial Key Laboratory for Computational Science and Material Design, Southern University of Science and Technology, Shenzhen 518055, China. ✉email: xgwan@nju.edu.cn; liuqh@sustech.edu.cn

Spin-orbit coupling (SOC) is the core of many emerging phenomena, including magnetocrystalline anisotropy[1], non-collinear magnetism[2–4], anomalous Hall effect[5], spin Hall effect[6], spin Seebeck effect[7,8], and Rashba–Edelstein effect[9]. These phenomena revolutionized and prospered various subfields in condensed matter such as spintronics[10,11], spin-orbitronics[12], and topological physics[13–15]. For instance, recently synthesized two-dimensional (2D) materials with long-range ferromagnetic order[16–18], lifting the Mermin–Wagner restriction[19], are stabilized via SOC-induced magnetocrystalline anisotropy. Similarly, SOC stabilizes the topological phases against the lattice distortion and thermal fluctuation by a nontrivial bandgap, which determines the realization temperature of the topological phases such as the quantum anomalous Hall (QAH) effect[20–22] and the quantum spin Hall effect[13,14].

However, designing materials with strong SOC in realistic materials has been quite straightforward because it simply links to the atomic mass of the constituent elements. Therefore, the candidates for spin-orbit active materials have mostly been limited to solids with heavy elements such as Bi, Sb, Te, Hg, Pt, and Pb[23–26]. Unfortunately, compounds containing heavy atoms usually have weaker chemical bonding; thus, they accommodate more native defects[27], leading to poor stability for performing exotic functionalities. A famous example is the topological insulator $Bi_2Te_3$ with bulk conductivity owing to the heavily $n$-type self-doping[28,29].

Therefore, it is crucial to thoroughly explore the potential of the SOC effect in materials with lighter elements. However, it is generally believed that SOC does not play an essential role in $3d$ transition metal materials, which are ideal for studying the interplay between symmetry, electronic occupation, and electron correlation[30]. On the other hand, the SOC effect was found comparable to the correlation in $4d$ and $5d$ series, leading to emergent quantum phases such as Weyl semimetal[31], topological Mott insulator[32,33], and quantum spin liquid[34,35]. The SOC effect of these systems was found to be more prominent in the presence of electron correlation, attributed to the electron localization induced by Coulomb repulsion that reduces the kinetic energy[33,34]. Recently, the cooperative effect between SOC and correlation was considered to explain the Fermi surface puzzle of the paramagnetic Fermi liquid $Sr_2RhO_4$[36], $Sr_2RuO_4$[37,38], as well as relatively large band splitting in other $4d$, $5d$, and $5f$ compounds[39,40]. These studies revealed the essential role of total angular momentum for the cooperative effects between SOC and correlation.

In this study, we aim to theoretically design materials with light elements but large effective SOC strength based on orbital symmetry, electron occupation, and the cooperative effect with correlation. The focus is on transition-metal magnetic materials, especially the $3d$ series, where the SOC strength is significantly smaller than the typical spin-exchange splitting. We propose that the cooperative effect of the electron correlation can significantly enhance the effective SOC through orbital polarization when there are partially occupied orbital multiplets around the Fermi level. Thereafter, we provide design principles and comprehensive databases, where we list all the Wyckoff positions and site symmetries that allow orbital multiplets in periodic crystals. The results indicate that 32 out of 80 layer groups and 125 out of 230 space groups can support large SOC effect. Therefore, for materials no matter recorded in existing databases or designed artificially, one can easily resort to our symmetry principles to predict promising candidates with strong effective SOC.

2D materials, specifically 2D magnets, have attracted significant attention because of their engineerable and integrable nature for future devices. Particularly, the high-temperature QAH effect has been investigated for the potential application of dissipationless electronics; however, it is challenging to realize[41]. Hence, we applied our procedure to Computational 2D materials database (C2DB)[42,43] and screened out 71 2D materials (from ~1600 candidates) with an orbital multiplet near the Fermi energy, enhancing the SOC effect. As opposed to the previous case-by-case search approach, we systematically obtained nine high-temperature QAH insulators with large nontrivial band gaps of hundreds of meV. Additionally, our symmetry principles and material candidates for enhanced SOC effect are valid for searching materials with strong magnetocrystalline anisotropy, which has a significant influence on industrial ferromagnetic materials with ultrahigh coercive fields. Our study paves a new avenue for realizing light-element materials with strong effective SOC for next-generation functional materials and devices in various fields.

## Enhancing SOC self-consistently by correlation

Here, we summarize the main idea of designing a large SOC effect in light $3d$ transition metal ions. First, we consider orbital multiplets to activate the first-order perturbation of SOC, i.e., the on-site term[44]. The presence of SOC splits the orbital degeneracy and slightly unquenches the orbital angular momentum, leading to orbital polarization. When the orbital multiplet is partially occupied, the strong on-site Coulomb correlation enhances the orbital polarization as well as the effective SOC. Because of the competition between correlation and hopping in $3d$ systems, the effective SOC is enhanced via the orbital polarization self-consistently. The mechanism is schematically shown in Fig. 1, and is presented in the following.

For a single $d$-shell ion exposed to the crystal field, only four types of orbital multiplets are allowed by the crystallographic symmetries, including three doublets, $E_1 = \{d_{xz}, d_{yz}\}$, $E_2 = \{d_{xy}, d_{x^2-y^2}\}$, $E_3 = \{d_{z^2}, d_{x^2-y^2}\}$, and one triplet, $T = \{d_{xy}, d_{yz}, d_{xz}\}$. Among them, the first-order SOC effect is absent in $E_3$. Herein, we demonstrate the physics using the $E_1$ doublet, whereas $E_2$ and $T$ multiplets can be found in Supplementary Note 1. As shown in Fig. 1a, the orbital angular momentum of the orbital doublet is quenched without SOC. Turning on the SOC, the on-site SOC Hamiltonian reads

$$\hat{H}_{SOC} = \sum_{m,m',\sigma,\sigma'} \lambda \langle m, \sigma | \mathbf{L} \cdot \mathbf{S} | m', \sigma' \rangle \hat{C}^{\dagger}_{m\sigma} \hat{C}_{m'\sigma'}, \quad (1)$$

where $\mathbf{L}$ and $\mathbf{S}$ are the angular momentum and spin operators, respectively; $\hat{C}^{\dagger}_{m\sigma}$ and $\hat{C}_{m\sigma}$ are the creation and annihilation operators on the electron state with orbital $m$; spin $\sigma$, and $\lambda$ denotes the strength of SOC. Because spin splitting typically overwhelms the SOC effect in $3d$ systems, we treat SOC as a perturbation with the spin-conserved part, $L_z S_z$, only, where the matrix representation is diagonal on the basis of the projected orbital angular momentum along the z-axis, i.e., $|l = 2, m = \pm 1\rangle = (|d_{xz}\rangle \pm i|d_{yz}\rangle)/\sqrt{2}$. Under such circumstances, the SOC Hamiltonian can be rewritten as $\hat{H}_{SOC} \approx \lambda \hbar \hat{L}_z / 2$, where $\hat{L}_z = \hbar(\hat{n}_{+1} - \hat{n}_{-1})$ is the operator of the orbital angular momentum and $\hat{n}_{\pm 1} = \hat{C}^{\dagger}_{\pm 1} \hat{C}_{\pm 1}$ is the occupation operator. In the single-ion limit, $\hat{H}_{SOC}$ splits the degenerated levels with a gap

$$\Delta E_0 = \lambda \hbar |\bar{L}_z| = \hbar^2 \lambda, \quad (2)$$

where $|\bar{L}_z| = \sum_{m \in occ.} |\langle m | \hat{L}_z | m \rangle| = \hbar$ is the expectation of $\hat{L}_z$ over the occupied state, i.e., orbital polarization. As shown in Fig. 1b, the splitting energy, $\Delta E_0$, in the $3d$ series is only a few dozens of meV owing to the light atomic mass.

In periodic solids, the inter-ions hopping broadens the energy levels of the atomic limit to form Bloch energy bands with a bandwidth proportional to the hopping integrals typically an order larger than $\lambda$. Consequently, for a $3d$ state with small SOC,

the splitting bands with opposite angular momenta lead to a slightly unquenched orbital polarization (Fig. 1c). However, the SOC-induced energy gap and orbital polarization can be iteratively enhanced by considering the electron correlation effect between different orbitals, i.e., $\hat{H}_C = U_{\text{eff}}\hat{n}_{+1}\hat{n}_{-1}$, where $U_{\text{eff}} = U - 3J$ for $d$ shell; $U$ and $J$ are the Coulomb repulsion and Hund coupling parameters, respectively[45]. When the orbital doublet is half-filled, $\hat{H}_C$ modifies the effective SOC and thus the energy gap at the mean-field level

$$\Delta E_{\text{eff}} = \Delta E_0 + U_{\text{eff}}\frac{|\bar{L}_z|}{\hbar}. \tag{3}$$

The derivation of Eq. (3) is provided in Supplementary Note 1. Starting from a tiny $\bar{L}_z$ and $\Delta E_0$, when considering $U_{\text{eff}}$, the separation between $L_z = \pm 1$ states enhances $|\bar{L}_z|$, which gives rise to a larger $\Delta E_{\text{eff}}$. In response, an enhanced $\Delta E_{\text{eff}}$ reduces the overlap of different orbitals, leading to an enlarged $|\bar{L}_z|$. As a result, the final orbital polarization is iteratively enlarged and settled self-consistently (Supplementary Note 1), as shown in Fig. 1d, and so is the energy gap.

Notably, a similar correlation-enhanced SOC effect was first revealed in the paramagnetic $4d$ transition-metal oxide $Sr_2RhO_4$ using a mean-field approach[36] when both the SOC and Coulomb terms involve the occupation difference between the total angular momentum, $|m_j| = 3/2$ and $|m_j| = 1/2$ states. In comparison, we apply the orbital polarization scheme to $3d$ transition systems, which typically have a ferromagnetic ground state with a weak SOC. Compared with the paramagnetic case, the enhancement of SOC in the ferromagnetic system is a function of $U - 3J$ instead of $U - J$. The enhancement of SOC can result in reinforcement of various emerging phenomena, as shown later.

## Materials design for correlation-enhanced SOC effect

Based on the mechanism described above, we extract the crucial principles to facilitate the design of light-element materials with large effective SOC. The $3d$ transition-metal elements of the materials should reside at the well-chosen Wyckoff positions, of which the site symmetries should permit the existence of orbital multiplets $E_1$, $E_2$, or $T$. Doublets $E_1$ and $E_2$ are allowed in both tetragonal, trigonal, and hexagonal point groups, whereas the triplet $T$ only exists in the cubic point groups, leading to 24 single point groups (Supplementary Table 1). The design principles lead to a comprehensive database with all the Wyckoff positions and

site symmetries, allowing orbital multiplets in 2D and 3D periodic crystals. The results indicate that 32 out of 80 layer groups and 125 out of 230 space groups can support the correlation-enhanced SOC effect, as listed in Supplementary Tables 2 and 3, respectively. Therefore, for materials no matter recorded in existing databases such as C2DB[42] and ICSD[46] or artificially designed, one can easily predict candidates with potentially strong SOC effect using our database (Supplementary Tables 2 and 3): (i) if the compound belongs to the required layer/space groups, and ii) if the transition-metal ion sits on the required Wyckoff positions.

For instance, we consider a $3d$ compound with space group P4$_2$/mmc (No. 131), which is one of the 125 groups. Although it has 18 different types of Wyckoff positions, as shown in Supplementary Table 2, the orbital doublet is permitted for enhanced SOC effect only when the transition-metal ion is located at Wyckoff positions 2e or 2f with the same site point group $-4$ m2. Notably, these symmetry requirements are established under the framework of nonmagnetic groups based on the assumption of strong Hund's exchange interaction in the $3d$ series. A similar methodology can be extended directly to the magnetic point and space groups.

To explicitly demonstrate the capability of our material design principle, we focus on a complete search of 2D light-element materials with strong effective SOC. The procedure is presented in Fig. 2. Starting from ~1600 2D materials in C2DB[42,43], we selected 859 entries with a set of $3d$ and $4d$ elements. Among them, by comparing the layer group each compound belongs to and the Wyckoff positions the transition-metal ions are located using Supplementary Table 3, we obtained 542 candidates. Thereafter, we perform the density functional theory (DFT) calculations to narrow the material pool with potentially correlation-enhanced SOC by examining if the considered orbital multiplets are partially occupied within 1.0 eV around the Fermi level. The calculation is based on a simple SOC-free framework with a spin-polarized generalized gradient approximation (GGA) exchange-correlation functional. Because the central principle is based on symmetry requirements, our results from such an economic approach are not sensitive to details such as the lattice constant, magnetic ordering configuration, and the choice of $U$. Finally, we pinned down the "best of class" of 71 material candidates (Table 1 and Supplementary Table 4), for which we perform more careful GGA+U+SOC calculations. According to Eq. (3), the enhanced effective SOC is typically one order larger than the original SOC strength of a dozen meV scale in $3d/4d$ systems.

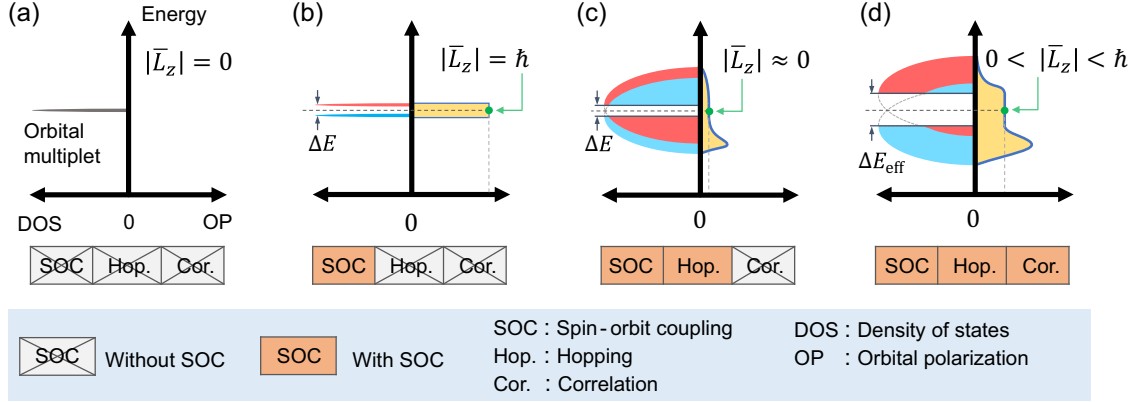

**Fig. 1 Schematic of boosting the effective SOC by correlation.** The density of states (DOS, the left part of the horizontal axis) and orbital polarization (OP, the right part of the horizontal axis) of the orbital doublet in single-ion limit (**a**, **b**) and periodic solids (**c**, **d**) considering the effects of SOC, inter-ion hopping and correlation. **a** Without SOC, the doubly degenerated orbital multiplet has quenched orbital angular momentum, $\bar{L}_z = 0$. **b** SOC splits the degenerated levels (denoted by red and blue colors) with an energy gap, $\Delta E$, and orbital polarization $|\bar{L}_z| = 1$ at half-filling. **c** The hopping between ions extends the energy levels into energy bands, leaving a tiny $\bar{L}_z$ because the SOC is typically significantly weaker than the inter-ion hopping. **d** Competition between the delocalized hopping and the on-site correlation effect significantly enhances $\bar{L}_z$ self-consistently, as well as the SOC effect and the energy gap.

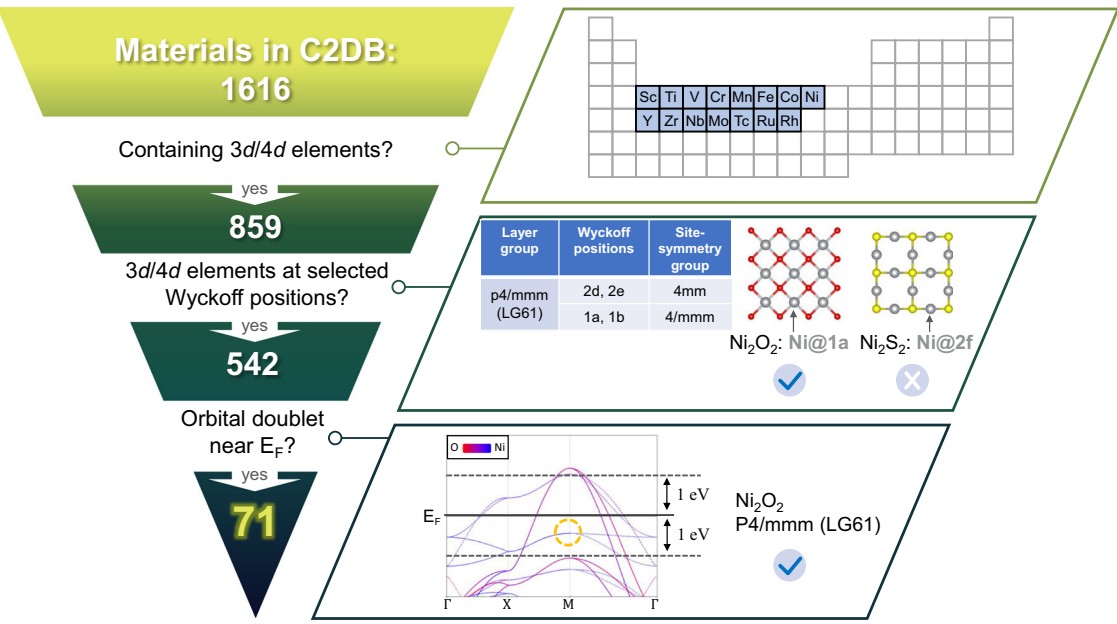

**Fig. 2 Design procedure of the 2D materials with enhanced SOC effect.** First, the elements of the $3d/4d$ series were chosen as $3d$: Sc~Ni and $4d$: Y~Rh, leading to 859 entries in C2DB. Second, the $3d/4d$ transition-metal ions need to reside at the Wyckoff positions whose site symmetries permit the existence of orbital multiplets. For instance, in a 2D $Ni_2O_2$ with layer group p4/mmm (LG 61), the Ni ion occupying 1a Wyckoff position Ni@1a (site point group 4/mmm) permits orbital doublet, whereas that in $Ni_2S_2$ Ni@2c (site point group mmm) does not. This filter obtains 542 entries remaining. Third, the considered orbital multiplets are partially occupied within 1.0 eV around the Fermi level, leading to 71 material candidates as the "best of class".

**Table 1 Abstract of the 71 2D material candidates with $3d/4d$ elements at selected Wyckoff positions with enhanced SOC effect.**

| Layer groups | Wyckoff positions (site-symmetry) | Representative compounds |
|---|---|---|
| p-4m2 (LG59) | 1a, 1b (−42m) | $CrS_2$ |
| p4/mmm (LG61) | 1a, 1b (4/mmm); 2d, 2e (4 mm) | $Ni_2O_2$ |
| p4/mbm (LG63) | 2a (4/m); 4c (4) | $Cr_2S_4$ |
| p4/nmm (LG64) | 2a (−42m); 2b (4 mm) | **$Fe_2Br_2$** |
| p3m1 (LG69) | 1a, 1b, 1c (3m) | $KTiS_2$ |
| p-31m (LG71) | 1a (−3m); 2b (32); 2c (3 m); 4e (3) | **$V_2Cl_6$, $V_2Br_6$, $V_2I_6$, $Fe_2Cl_6$, $Fe_2I_6$, $RuI_3$** |
| p-3m1 (LG72) | 1a (−3m); 2b, 2c (3m) | **$Fe_2S_2$, $Fe_2Se_2$** |
| p-6m2 (LG78) | 1a, 1b, 1c (−6m2); 2d, 2e, 2 f (3 m) | $FeCl_2$ |
| p-62m (LG79) | 1a (−62m); 2b (−6); 2c (3 m); 4e (3) | $Ti_2Br_6$ |

Nine material candidates of high-temperature quantum anomalous Hall insulators are marked bold.

As a bonus, we systematically obtain nine materials with nontrivial SOC-induced bandgaps (Table 1), rendering them large-gap QAH insulators against thermal excitation[21] and local disorder[47]. Although some materials in Table 1 and certain artificial structures are occasionally predicted as large-gap QAH insulators[48–57], a comprehensive understanding of the nontrivial gaps with temperatures above the room temperature as well as systematic material search are still lacking. Conversely, the large-gap QAH insulators in our framework can be well understood and exhaustively obtained from our material candidates (Supplementary Table 4) by performing more delicate GGA+U Wannier-representation tight-binding calculations[58–61] (see Methods). In the following, we consider the monolayers of honeycomb transition-metal monochalcogenides, $Fe_2X_2$ (X = S, Se), to demonstrate our theory.

## QAH insulator with huge nontrivial gap

Monolayer $Fe_2X_2$, which is shown in Fig. 3a as two stacking honeycomb FeX sublayers, has layer group p-3 m1 (No. 72) with each Fe atom of $Fe_2X_2$ at the 2c Wyckoff position (site symmetry group 3 m). According to Supplementary Table 3, three types (1a, 2b, and 2c) of the Wyckoff positions support orbital multiplets, rendering $Fe_2X_2$ a potential candidate for large effective SOC. The $\{d_{xz}, d_{yz}\}$ orbitals of Fe ions form the basis functions of the 2D irreducible representation at the $K$ and $K'$ valleys at the Fermi level. Hence, there is a Dirac cone with linear dispersions at each valley owing to the constraint of the corresponding little-group[62]. Our spin-polarized GGA calculations show that the Dirac cones are half-occupied under the ferromagnetic ground state with a magnetic moment of 4 $\mu_B$/Fe (Fig. 3b and Supplementary Note 4). Turning on SOC, uniaxial magnetic anisotropy with out-of-plane moments is preferred as discussed later.

Through our procedure, the SOC gap is significantly enhanced by the correlation effect self-consistently in $Fe_2X_2$. Considering $Fe_2S_2$, the bandgap opened solely by the first-order SOC could be obtained by either treating SOC as a perturbation with a given $U$ value or by setting $U = 0$ in a self-consistent calculation (Supplementary Note 4). Both approaches yield a typical first-order SOC gap of approximately 43 meV, which is large for Fe with SOC strength of ~15 meV[63]. When we consider the correlation effect self-consistently, the SOC gap is enhanced significantly, reaching 628 meV at the $K$ valley ($U = 3$ eV), as shown in Fig. 3c. The enhancement $\Delta E_{eff}/\Delta E_0$ is qualitatively consistent with the analytical results, which reveals the dependence of $U_{eff} = U − 3J$ (Supplementary Note 4). By checking the Chern number, $C = 1$, and the chiral edge modes (Fig. 3d), we efficiently predict that monolayer $Fe_2S_2$ is a potential candidate for high-temperature QAH insulators with a large nontrivial gap. Similarly, the

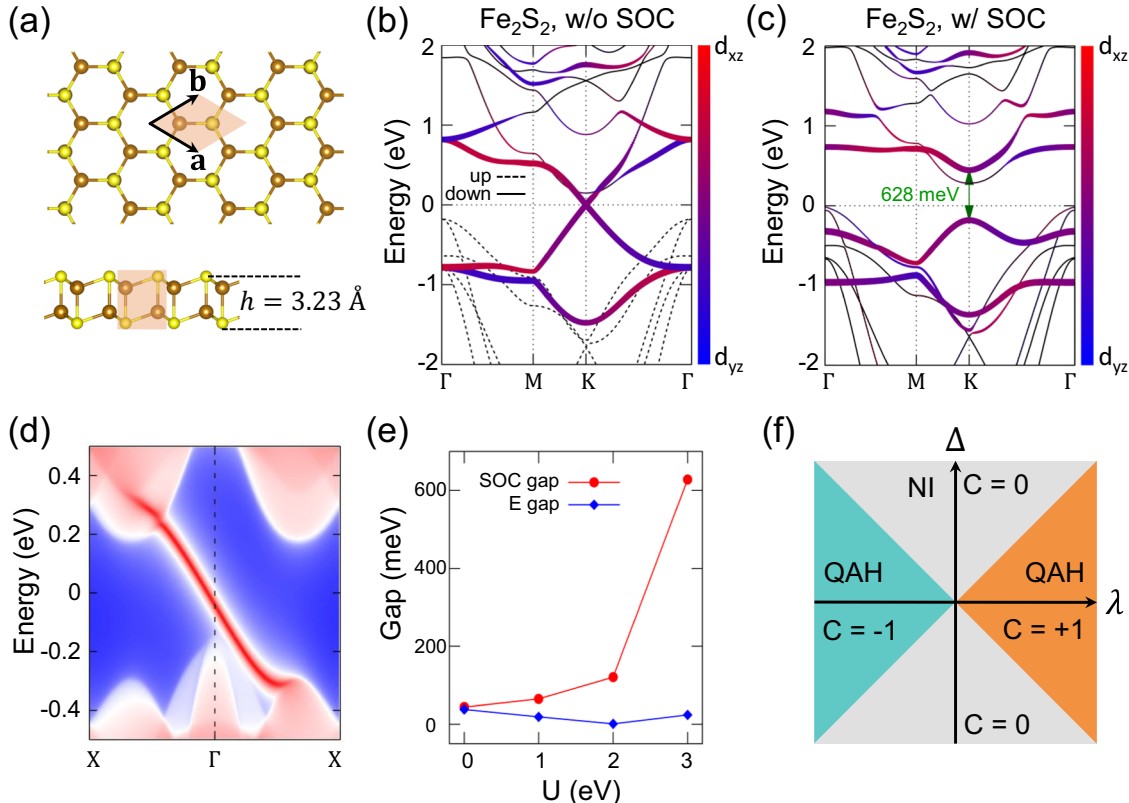

**Fig. 3 Fe₂X₂ (X = S, Se) monolayer. a** Sketch of the honeycomb lattice. **b, c** Band structure of the Fe₂S₂ crystal without and with spin-orbit coupling with $U = 3$ eV, with color mapping the projection on $\{d_{xz}, d_{yz}\}$ orbitals. Here, we consider the global coordinate with the crystal $c$-axis along the z-axis. **d** Gapless edge state inside the large bulk gap window of 400 meV. **e** Evolution of the SOC gap and the $E$-field induced gap as a function of $U$ in DFT calculation. **f** Haldane's phase diagram of transition between the normal insulator (NI) and the quantum anomalous Hall (QAH) insulator obtained from the corresponding tight-binding model.

correlation-enhanced nontrivial gap in monolayer honeycomb Fe₂Se₂ reaches 798 meV at the $K$ valley, as shown in Supplementary Note 4.

Notably, the Dirac cones of Fe₂X₂ can also be gapped in a trivial way by an out-of-plane external electric field owing to the onsite energy splitting of different $d$ electrons at two FeX sublayers. Because the electric field could not effectively separate the orbital degeneracy of $\{d_{xz}, d_{yz}\}$, the correlation effect can hardly enlarge the gap. Therefore, the electric-field-induced gap is more insensitive with increasing $U$ (Fig. 3e). In addition, the competition between the SOC-induced nontrivial gap and the electric field-induced trivial gap in monolayer Fe₂X₂ is analogous to the Haldane phase diagram[64] (Fig. 3f), which is quite rare in realistic materials. More detailed calculation results for Fe₂X₂ are presented in Supplementary Note 4.

## Discussion

The main results of our study are based on the assumption of strong exchange splitting, for which the interaction between electrons with parallel spins dominates. In principle, the correlation-enhanced SOC effect should also be valid under more general circumstances. In moderate spin-splitting case, considering a general Kanamori-type correlation (Supplementary Note 5), we find that although spin flipping process weakens the orbital polarization of each spin channel, including the contribution of the opposite spin channels does not qualitatively change the SOC enhancement. Even for the spin-degenerate case, correlation can generally enhance the SOC effect. As reported in Liu et al.[36], SOC splits the degenerated states with different total

angular momentum $|m_j| = 3/2$ and $|m_j| = 1/2$. The resultant "spin-orbit polarization", i.e., the occupation difference between $|m_j|$ states, plays an essential role in the enhanced SOC effect along with the correlation between parallel and antiparallel spins. Therefore, our results together with previous studies[36–40] demonstrate that correlation can in general enhance SOC via different polarization effects in various magnetic states.

In addition to the large-gap QAH insulators, the paradigm we used to design materials with large SOC effects could be applied to various scenarios. For instance, the long-range magnetism of 2D magnetic materials can only be stabilized by uniaxial anisotropy[1]. The single-ion magnetocrystalline anisotropy, which is one of the main components of uniaxial anisotropy, originates from the SOC effect[65]. With the reinforcement of the orbital polarization and SOC effect, the magnetocrystalline anisotropic energy, $E_{MAE}$, for 2D magnetic materials with $\{d_{xz}, d_{yz}\}$ frontier orbital doublet can significantly increase as[66]

$$E_{MAE} = \frac{1}{2}\left(\hbar^2\lambda + U_{\text{eff}}\frac{|\bar{L}_z|}{\hbar}\right)|\cos\theta|, \qquad (4)$$

where $\theta$ is the angle between the magnetic moment and the vector normal to the 2D plane (Supplementary Note 16). This boosted magnetic anisotropy can stabilize the magnetic order in 2D magnetic materials with out-of-plane magnetic moments, getting rid of the Mermin–Wagner theorem for isotropic spins. The DFT calculations on the magnetocrystalline anisotropy of various materials yield qualitatively consistent results, as shown in Supplementary Note 4.

To summarize, we provide design principles for large SOC effects with the help of orbital degeneracy, electron occupation, and correlation, eliminating the need for heavy elements. To activate the correlation-enhanced SOC effect, we combined symmetry analysis of the transition-metal sites residing at specific Wyckoff positions and first-principles calculations to examine the partially occupied orbital multiplets around the Fermi level. Applying the guiding principles to the C2DB database, we found 71 2D material candidates supporting the correlation-enhanced SOC effect and nine compounds as potential candidates for high-temperature QAH insulators. The procedure can be easily extended for designing and searching 3D light-element materials with strong effective SOC in various fields of condensed matter physics, such as spintronics, spin-orbitronics, and topological phases of matter.

## Methods

**First-principles calculations.** First-principles calculations were based on DFT with generalized gradient approximation (GGA)[67,68] for exchange correlation potential. The Perdew−Burke−Ernzerholf (PBE) functional was used for the GGA as implemented in Vienna ab initio simulation package (VASP)[69]. The electron-ion interaction was treated using projector-augmented-wave (PAW) potentials[70] with a planewave-basis cutoff of 500 eV. The entire Brillouin zone was sampled using the Monkhorst−Pack[71] method. A vacuum of 15 Å was used to avoid artificial interactions caused by the periodic boundary conditions. Because of the correlation effects of 3d electrons in Fe atoms, we employed the GGA + U approach within the Dudarev scheme[72] and set U varying from 0 to 3 eV. Both $Fe_2S_2$ and $Fe_2Se_2$ were fully relaxed until the force on each atom was less than 0.01 eV/Å, and the total energy minimization was performed with a tolerance of $10^{-5}$ eV. The Wannier representation was constructed by projecting the Bloch states from the first-principles calculations to Fe-3d, S-3p, and Se-4p orbitals[58–60]. The edge states and Berry curvature were calculated in the tight-binding models constructed using the Wannier representation as implemented in the WannierTools package[61]. The global symmetries of each material and the site symmetries of the Wyckoff positions during the screening procedure were examined via the FINDSYM module[73] of the ISOTROPY software.

## Data availability

The supportive data for the findings in this study are available from the corresponding authors upon reasonable request.

## Code availability

The computation code for getting the theoretical prediction is available from the corresponding authors upon reasonable request.

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

## Acknowledgements

This work was supported by the National Key R&D Program of China under Grant Nos. 2020YFA0308900, 2017YFA0303203, and 2018YFA0305704; the National Natural Science Foundation of China under Grant Nos. 11874195, 12188101, 51721001, and 11790311; Guangdong Innovative and Entrepreneurial Research Team Program under Grant No. 2017ZT07C062; Guangdong Provincial Key Laboratory for Computational Science and Material Design under Grant No. 2019B030301001; Science, Technology and Innovation Commission of Shenzhen Municipality (No. ZDSYS20190902092905285); Center for Computational Science and Engineering of Southern University of Science and Technology. X.W. also acknowledges the support from the Tencent Foundation through the XPLORER PRIZE.

## Author contributions

Q.L. and X.W. supervised the whole project. L.W., J.L., X.W., and Q.L performed the symmetry analysis. Q.Y. and B.G. performed DFT calculations. Z.H., J.L., and L.W. carried out the materials screening. J.L. and Q.L. prepared the manuscript with contributions from all authors. J.L., Q.Y., and L.W. contributed equally to this work.

## Competing interests

The authors declare no competing interests.
