## [Peer Review File · Nature Communications]

Reviewers' Comments:

Reviewer #1:

Remarks to the Author:

The paper points out that robust topological properties that are traditionally sought in systems with heavy elements could also be realised in systems with 3d and 4d transition metal ions. This (to my knowledge original) idea is rationalised by the fact that the Coulomb interaction enhances the spin-orbit coupling (SOC) strength. This has been known in other context (paramagnetic metals Sr₂RhO₄ and ruthenates), but has not been stressed for topological insulators. The paper considers ferromagnetic monolayers as a realisation of the physics, browse through the corresponding material database and identify the key material properties and also few candidates that would reveal proposed physics. For Fe₂S₂ monolayer they perform explicit LDA+U calculations and demonstrate how SOC and Coulomb interaction collaborate to open up a broad gap.

I judge the finding of the authors and its demonstration on monolayer examples is important both for the topological insulator community and for the correlated electron community and find it might merit the publication.

The discussion should be improved in following aspects:

1. previous work on Coulomb interaction enhancement of SOC should be discussed better. Namely, original reference Liu, ..., and Andersen is cited, but its importance should be highlighted more. Some more recent works, J. Phys.: Condens. Matter 32 (2020) 445601, Phys. Rev. Lett. 116, 106402, Phys. Rev. Lett. 120, 126401, could also be cited. But in particular, the relation of the perturbative argument by the authors, with the crucial difference being that here ferromagnetism is assumed, to the earlier work should be clearly discussed.
2. For the perturbative argument the discussion should be improved, the authors mention how orbital polarisation is iteratively increased but it is not clear which iterations they have in mind. In Liu, ..., and Andersen, and in J. Phys.: Condens. Matter 32 (2020) 445601 the enhanced value of SOC is expressed in terms of a susceptibility, and perhaps authors can get inspiration from there to make the discussion of the enhancement more transparent.
3. English should be improved.

Reviewer #2:

Remarks to the Author:

The authors present in their manuscript an interesting principle of how to search for materials with large effective SOC. This idea is not entirely new, but what is new is to put this idea into a solid basis of group theoretical analysis, including the identification of promising candidates showing large effective SOC without involving heavy and expensive 5d elements.

In that respect, I think that the paper could be interesting for a broad range of researchers, both working on theoretical modelling and experimental realisations of spin-orbit coupled materials. However, I have some troubles with parts of the manuscript, in particular the analytic treatment of the effective SOC.

1) My main worry is the heavily oversimplified interaction Hamiltonian, as for instance introduced in the supplementary material, page 3 line 59. The authors write that the on-site correlation effect is captured by the interaction between parallel spins in different orbitals, and the prefactor of this term is U. Now this statement, if at all, applies only to the fully spin-polarised insulating situation, where both charge and spin fluctuations are basically zero, and the mean-field treatment gives more or less the correct answer. But that means that this whole procedure and argument is only valid in exactly this case, i.e. one spin channel has to be completely empty. Now this is a very strong limitation, which is not stated in the manuscript.

There are two ways out: First, since the authors are talking about orbital multiplets, introduce a proper multiplet hamiltonian of Slater or Kanamori type and try to do a similar analysis - which is much more involved since there are at least two important parameters, Hubbard U and Hund's coupling J. I think that a proper treatment of effects of Hund's coupling are worthwhile. Second, put the proper limitations of the arguments and analytic insights into the paper.

By the way, exactly this argument of enhanced SOC has been put forward already quite early on in this work:

Liu et al. Phys.Rev.Lett 101 026408 (2008)

They develop also a mean-field theory for a doublet system, but based on an appropriate Hamiltonian. In the present work, this needs also to be done!

2) There is a lot of analytic treatment in the paper, but the authors do not try to connect the obtained formulas for the effective band splitting, or effective SOC as they call it, to the actual GGA+U calculations. There is a U parameter in the calculations, so it should be easy to check whether the analytic formulas work or not. The authors quote very precise values for the enhanced gap. Is this consistent with formula (3)? If this does not fit together, there is a problem.

2) A bit related to 1): The DFT calculations are done for materials with strong spin splitting, and we know that in those systems the effect of including the interaction through a GGA+U treatment is strongest. Without spin-splitting, the effect of this mean-field treatment is only minute. Does this again mean that this proposal of enhanced SOC is working only for strongly magnetic materials? What is the situation for non-magnetic materials, or above some magnetic ordering temperature? Certainly, the last question cannot be answered exactly by DFT, but one can do non-magnetic DFT calculations and argue that this should be not too far from a high-temperature solution, at least what concerns the magnetic ordering.

3) One smaller thing that I noticed: In Fig. 2 and the related discussion in the text, Ni₂O₂ is discussed a lot, but then the band structure in Figure 2 is calculated for FeO₂. Why?

Reviewer #3:

None

Response to Referee's comments (NCOMMS-21-29249-T)

“Designing light-element materials with large effective spin-orbit coupling” by Jiayu Li et al.

We sincerely thank both referees for taking their time to carefully review our work and providing insightful comments that indeed helped us to improve it. We are happy to note that we have carefully addressed all the referees' comments. The itemized response is provided in the following. Note that the original referee reports are in blue italic font, and the corresponding main-text changes are shown in italic font here and are highlighted in the revised manuscript.

Comments of Referee #1 and authors' reply:

1.1 Referee: The paper points out that robust topological properties that are traditionally sought in systems with heavy elements could also be realised in systems with 3d and 4d transition metal ions. This (to my knowledge original) idea is rationalised by the fact that the Coulomb interaction enhances the spin-orbit coupling (SOC) strength. This has been known in other context (paramagnetic metals Sr₂RhO₄ and ruthenates), but has not been stressed for topological insulators. The paper considers ferromagnetic monolayers as a realisation of the physics, browse through the corresponding material database and identify the key material properties and also few candidates that would reveal proposed physics. For FeS₂ monolayer they perform explicit LDA+U calculations and demonstrate how SOC and Coulomb interaction collaborate to open up a broad gap.

I judge the finding of the authors and its demonstration on monolayer examples is important both for the topological insulator community and for the correlated electron community and find it might merit the publication.

The discussion should be improved in following aspects:

Reply: We thank the referee for appreciating the importance of the work and the recommendation for publication.

1.2 Referee: 1. previous work on Coulomb intereaction enhancement of SOC should be discussed better. Namely, original reference Liu,..., and Andersen is cited, but its importance should be highlighted more. Some more recent works, J. Phys.: Condens. Matter 32 (2020) 445601, Phys. Rev. Lett. 116, 106402, Phys. Rev. Lett. 120, 126401, could also be cited. But in particular, the relation of the perturbative argument by the authors, with the crucial difference being that here ferromagnetism is assumed, to the earlier work should be clearly discussed.

Reply: We thank the referee for raising this point. In the revised manuscript, we clearly point out the originality of Liu et. al. on proposing the correlation-enhanced SOC effect in paramagnetic system and emphasize that our results are based on ferromagnetic systems. In addition, we add the papers pointed out by the referee as well as the other relevant references. The new text added are in the following:

Page 4 Paragraph 1: *“Recently, the cooperative effect between SOC and correlation was considered to explain the Fermi surface puzzle of the paramagnetic Fermi liquid Sr_2RhO_4 ³⁶, Sr_2RuO_4 ^{37, 38}, as well as relatively large band splitting in other 4d, 5d, and 5f compounds^{39, 40}. These studies revealed the essential role of total angular momentum for the cooperative effects between SOC and correlation.”*

Page 8 Paragraph 2: *“Notably, a similar correlation-enhanced SOC effect was first revealed in the paramagnetic 4d transition-metal oxide Sr_2RhO_4 using a mean-field approach³⁶ when both the SOC and Coulomb terms involve the occupation difference between the total angular momentum, $|m_j| = 3/2$ and $|m_j| = 1/2$ states. In comparison, we apply the orbital polarization scheme to 3d transition systems, which typically have a ferromagnetic ground state with a weak SOC. Compared with the paramagnetic case, the enhancement of SOC in the ferromagnetic system is a function of $U - 3J$ instead of $U - J$.”*

Page 13 Paragraph 1: *“Even for the spin-degenerate case, correlation can generally enhance the SOC effect. As reported in Ref. ³⁶, SOC splits the degenerated states with different total angular momentum $|m_j| = 3/2$ and $|m_j| = 1/2$. The resultant “spin-orbit polarization”, i.e., the occupation difference between $|m_j|$ states, plays an essential role in the enhanced SOC effect along with correlation between parallel and antiparallel spins. Therefore, our results together with previous studies³⁶⁻⁴⁰ demonstrate that correlation can in general enhance SOC via different polarization effects in various magnetic states.”*

[36] Liu et. al., Phys. Rev. Lett. 101, 026408 (2008).

[37] Zhang et. al. , Phys. Rev. Lett. 116, 106402 (2016).

[38] Kim et. al., Phys. Rev. Lett. 120, 126401 (2018).

[39] Triebel et. al., Phys. Rev. B 98, 205128 (2018).

[40] Riseborough et. al., J. Phys. Condens. Matter 32, 445601 (2020).

1.3 Referee: 2. For the perturbative argument the discussion should be improved, the authors mention how orbital polarisation is iteratively increased but it is not clear which iterations they have in mind. In Liu,..., and Andersen, and in J. Phys.: Condens. Matter 32 (2020) 445601 the enhanced value of SOC is expressed in terms of a susceptibility, and perhaps authors can get inspiration from there to make the discussion of the enhancement more transparent.

Reply: We thank the referee for the helpful suggestion. In Eq. (3) of the main text, the orbital polarization $|\bar{L}_z|$ originating from SOC is tiny before iteration. This is because inter-ions hopping

broadens the energy levels of distinct orbitals and nearly quenches the orbital polarization. When considering correlation, the effective gap ΔE_{eff} is then slightly enhanced. In response, a larger ΔE_{eff} enhances the orbital polarization $|\bar{L}_z|$ as the gap further reduces the overlap of different orbitals. Therefore, the orbital polarization is iteratively enhanced.

Furthermore, we note that the correlation-enhanced SOC effect is manifested even before the mean-field treatment. Still taking the orbital doublet as an example, the correlation term $\hat{H}_c = \frac{U_{\text{eff}}}{4} \left(\hat{n}^2 - \frac{\hat{L}_z^2}{\hbar^2} \right)$ (Eq. (5) in Supplementary Note 1) with $-\hat{L}_z^2$ directly provides an energy offset on states with opposite angular momenta. Thus, the splitting between single-particle energies of different l_z states is iteratively enlarged. The case of orbital triplet yields the same effect, as shown in Eq. (13) in Supplementary Note 1.

In the revised version, we clarify the iteration process in Page 7 Paragraph 2: “Starting from a tiny \bar{L}_z and ΔE_0 , when considering U_{eff} , the separation between $L_z = \pm 1$ states enhances $|\bar{L}_z|$, which gives rise to a larger ΔE_{eff} . In response, an enhanced ΔE_{eff} reduces the overlap of different orbitals, leading to an enlarged $|\bar{L}_z|$. As a result, the final orbital polarization is iteratively enlarged and settled self-consistently (Supplementary Note 1), as shown in Fig. 1(d), and so is the energy gap”.

Following the referee’s suggestion, we also express $|\bar{L}_z|$ as a linear response of λ , i.e., $|\bar{L}_z| = \chi\lambda$, where χ is the susceptibility [Phys. Rev. Lett. 101, 026408 (2008)]. The self-consistent solution of the orbital polarization satisfies the same linear dependence on the effective SOC parameter, extracted from the enhanced SOC gap $\Delta E_{\text{eff}} = \lambda_{\text{eff}}\hbar^2$. Finally, substituting $|\bar{L}_z^{sc}| = \chi\lambda_{\text{eff}}$ into Eq. (3) yields an enhancement of the energy gap as

$$\frac{\Delta E_{\text{eff}}}{\Delta E_0} \approx \left[1 - \frac{(U - 3J)\chi}{\hbar^3} \right]^{-1}. \quad (\text{R1})$$

Hence, the SOC splitting is enhanced as a function of $U - 3J$. We have added the above self-consistent results to Supplementary Note 1 (Page 4).

1.4 Referee: 3. English should be improved.

Reply: We have hired a professional team to polish the English writing throughout the paper.

 Comments of Referee #2 and authors’ reply:

2.1 Referee: The authors present in their manuscript an interesting principle of how to search for materials with large effective SOC. This idea is not entirely new, but what is new is to put this idea into a solid basis of group theoretical analysis, including the identification of promising candidates showing large effective SOC without involving heavy and expensive 5d elements.

In that respect, I think that the paper could be interesting for a broad range of researchers, both working on theoretical modelling and experimental realisations of spin-orbit coupled materials. However, I have some troubles with parts of the manuscript, in particular the analytic treatment of the effective SOC.

Reply: We thank the referee for appreciating the potential impact of the work.

2.2 Referee: 1) My main worry is the heavily oversimplified interaction Hamiltonian, as for instance introduced in the supplementary material, page 3 line 59. The authors write that the on-site correlation effect is captured by the interaction between parallel spins in different orbitals, and the prefactor of this term is U . Now this statement, if at all, applies only to the fully spin-polarised insulating situation, where both charge and spin fluctuations are basically zero, and the mean-field treatment gives more or less the correct answer. But that means that this whole procedure and argument is only valid in exactly this case, i.e. one spin channel has to be completely empty. Now this is a very strong limitation, which is not stated in the manuscript.

There are two ways out: First, since the authors are talking about orbital multiplets, introduce a proper multiplet hamiltonian of Slater or Kanamori type and try to do a similar analysis - which is much more involved since there are at least two important parameters, Hubbard U and Hunds coupling J . I think that a proper treatment of effects of Hunds coupling are worthwhile. Second, put the proper limitations of the arguments and analytic insights into the paper.

By the way, exactly this argument of enhanced SOC has been put forward already quite early on in this work:

Liu et al. Phys.Rev.Lett 101 026408 (2008)

They develop also a mean-field theory for a doublet system, but based on an appropriate Hamiltonian. In the present work, this needs also to be done!

Reply: We agree with the referee that Eq. (3) is derived by only considering correlation between parallel spin states. This is valid in strong spin-splitting $3d$ systems where the Hund coupling J significantly overwhelms SOC. In this case, we find that for d -orbital states, the effective interaction between parallel spin is $U_{\text{eff}} = U - 3J$ (U and J the Coulomb and Hund parameters, respectively). In the revised version, we replace the single U parameter by $U_{\text{eff}} = U - 3J$, and add the following discussion to the main text:

Page 7 Paragraph 2: “However, the SOC-induced energy gap and orbital polarization can be iteratively enhanced by considering the electron correlation effect between different orbitals, i.e., $\hat{H}_C = U_{eff}\hat{n}_{+1}\hat{n}_{-1}$, where $U_{eff} = U - 3J$ for d shell; U and J are the Coulomb repulsion and Hund coupling parameters, respectively⁴⁵”

[45] Kanamori, Prog. Theor. Phys. 30, 275-289 (1963).

On the other hand, we suggest that the correlation-enhanced SOC effect is prevalent in more general circumstances besides the strong spin splitting case, where other correlation terms should be included. Following the referee’s suggestion, we consider a more general Kanamori-type Hamiltonian for correlation effect. Taking orbital doublet $E_1 = \{d_{xz}, d_{yz}\}$ as example, the formula of band splitting for σ spin channel is written as

$$\Delta E_{\sigma}^K \approx \Delta E_{\text{eff}} + 2J \frac{|\bar{L}_{z,\bar{\sigma}}|}{\hbar} - \left| \frac{\phi_s}{\bar{n}_{\sigma} - \bar{n}_{\bar{\sigma}}} \right|^2 \Gamma(\bar{L}_{z,\sigma}, \bar{L}_{z,\bar{\sigma}}). \quad (\text{R2})$$

Besides the previously obtained gap ΔE_{eff} in Eq. (3), the orbital polarization of the opposite spin channel $\bar{L}_{z,\bar{\sigma}}$ and a spin-flipping-induced term $\propto \phi_s^2 = \langle \hat{C}_{m\sigma}^{\dagger} \hat{C}_{m\bar{\sigma}} \rangle^2$ appear, where $\bar{n}_{\sigma} - \bar{n}_{\bar{\sigma}}$ counts the exchange effect. With zero spin flipping $\phi_s = 0$, the energy gap is still enhanced even the opposite spin channel is considered. On the other hand, while the spin flipping term neglected in our previous version will reduce the enhancement, the effect of enhanced SOC still maintains in $3d$ systems with spin splitting. The details are provided in Supplementary Note 5.

In the revised manuscript, we add the following text in Page 12 Paragraph 3: “The main results of our study are based on the assumption of strong exchange splitting, for which the interaction between electrons with parallel spins dominates. In principle, the correlation-enhanced SOC effect should also be valid under more general circumstances. In moderate spin-splitting case, considering a general Kanamori-type correlation (Supplementary Note 5), we find that although spin flipping process weakens the orbital polarization of each spin channel, including the contribution of the opposite spin channels does not qualitatively change the SOC enhancement.”

2.3 Referee: 2) There is a lot of analytic treatment in the paper, but the authors do not try to connect the obtained formulas for the effective band splitting, or effective SOC as they call it, to the actual GGA+U calculations. There is a U parameter in the calculations, so it should be easy to check whether the analytic formulas work or not. The authors quote very precise values for the enhanced gap. Is this consistent with formula (3)? If this does not fit together, there is a problem.

Reply: We thank the referee for this helpful suggestion. As discussed in Reply #1.3, we express the enhancement of SOC gap self-consistently in terms of susceptibility χ that characterizes linear dependence of orbital polarization on the intrinsic SOC $|\bar{L}_z| = \chi\lambda$, as

$$\frac{\Delta E_{\text{eff}}}{\Delta E_0} \approx \left[1 - \frac{(U - 3J)\chi}{\hbar^3} \right]^{-1}. \quad (\text{R1})$$

Eq. (R1) indicates that the SOC increment varies as a function of $(U - 3J)$ with U and J the Coulomb and Hund parameters, respectively. Taking Fe_2S_2 , such a dependence is now examined by our GGA+U calculation, as shown in following figure:

Fig. R1. SOC increment as a function of the effective parameter $U_{\text{eff}} = U - 3J$. The red curve is obtained by the four-band tight-binding model of Fe_2S_2 with interaction between parallel spin states considered only, while the other three curves are obtained by GGA+U calculations with different U and J . Consistent tendency is shown from all results. As expected, the simplified correlation term is more accurate in a strong exchange case (large J), with a reasonable susceptibility $\chi/\hbar \approx 0.73 \text{ eV}^{-1}$ fitted by Eq. (R1) with $J = 1.0 \text{ eV}$.

The consistency between the analytical results and DFT calculations indicates that the correlation between d orbitals with parallel spin is essentially responsible for the SOC enhancement in ferromagnetic systems considered in our work. In the revised version, we add the examination of the comparison between analytical results and DFT calculations on Page 12 Paragraph 1: “The enhancement $\Delta E_{\text{eff}}/\Delta E_0$ is qualitatively consistent with the analytical results, which reveals the dependence of $U_{\text{eff}} = U - 3J$ (Supplementary Note 4).”

2.4 Referee: 2) A bit related to 1): The DFT calculations are done for materials with strong spin splitting, and we know that in those systems the effect of including the interaction through a GGA+U treatment is strongest. Without spin-splitting, the effect of this mean-field treatment is only minute. Does this again mean that this proposal of enhanced SOC is working only for strongly magnetic materials? What is the situation for non-magnetic materials, or above some magnetic ordering temperature? Certainly, the last question cannot be answered exactly by DFT, but one can do non-magnetic DFT calculations and argue that this should be not too far from a high-temperature solution, at least what concerns the magnetic ordering.

Reply: We thank the referee for the illuminating comment. The correlation-enhanced SOC effect is firstly proposed in the paramagnetic Sr_2RhO_4 by Liu et.al. [Phys. Rev. Lett. 101, 026408 (2008)] as pointed out by the referee in #2.2. In their original work, they found that correlation enhances the SOC-induced band splitting via the so-called **spin-orbit (SO) polarization** $\bar{p} = \bar{n}_{3/2} + \bar{n}_{-3/2} - \bar{n}_{1/2} - \bar{n}_{-1/2}$, as SOC splits the degeneracy of $|m_j| = 3/2$ and $|m_j| = 1/2$ states. The mechanism was also verified by DFT calculations under the framework of LDA+U+SOC. Therefore, the SO polarization serves as a pivot in spin-degenerate case, as the orbital polarization does in spin-splitting case. The difference between the two cases is that interaction between orbitals with opposite spin $U'_{\text{eff}}\hat{n}_{m,\uparrow}\hat{n}_{m',\downarrow}$ also matters in spin-degenerate system. Therefore, we agree with the referee that the major term considered in our spin-splitting case, i.e., $U_{\text{eff}}\hat{n}_{m,\uparrow}\hat{n}_{m',\uparrow}$, is not dominant in spin-degenerate systems. However, a unified description on the enhanced SOC effect in both cases can be done by considering the more general Kanamori-type Hamiltonian for correlation effect, as mentioned in Reply #2.2. Our results of Eqs. (3)-(5) together with previous studies demonstrate that correlation can in general enhance SOC via different polarization effects in various magnetic states.

As suggested by the referee, we perform nonmagnetic DFT+U calculations on the representative example monolayer Fe_2S_2 . As shown below, the SOC-induced band gap is also enhanced by U, probably because of the cooperative effect between correlation and other types of polarization.

Fig. R2. Correlation-enhanced SOC gaps in ferromagnetic (FM) and nonmagnetic (NM) Fe_2S_2 , where the increment in nonmagnetic case is less significant.

In the revised version, we add the following text in Page 13 Paragraph 1: “*Even for the spin-degenerate case, correlation can generally enhance the SOC effect. As reported in Ref. ³⁶, SOC splits the degenerated states with different total angular momentum $|m_j| = 3/2$ and $|m_j| = 1/2$. The resultant “spin-orbit polarization”, i.e., the occupation difference between $|m_j|$ states, plays an essential role in the enhanced SOC effect along with correlation between parallel and antiparallel spins. Therefore, our results together with previous studies³⁶⁻⁴⁰ demonstrate that correlation can in general enhance SOC via different polarization effects in various magnetic states.*”

2.5 Referee: 3) *One smaller thing that I noticed: In Fig. 2 and the related discussion in the text, Ni_2O_2 is discussed a lot, but then the band structure in Figure 2 is calculated for FeO_2 . Why?*

Reply: We apologize for confusing the referee. We used the band structure of FeO_2 in the previous version because it looks cleaner, with pure Fe-*d* orbitals around the Fermi level. Now we have replaced the band structure of FeO_2 and uniformly take Ni_2O_2 as the example to illustrate the procedure shown in Fig. 2.

Figure 2. Design procedure of the 2D materials with enhanced SOC effect. First, the elements of the 3d/4d series were chosen as 3d: Sc~Ni and 4d: Y~Rh, leading to 859 entries in C2DB. Second, the 3d/4d transition-metal ions need to reside at the Wyckoff positions whose site symmetries permit the existence of orbital multiplets. For instance, in a 2D Ni₂O₂ with layer group p4/mmm (LG 61), the Ni ion occupying 1a Wyckoff position Ni@1a (site point group 4/mmm) permits orbital doublet, whereas that in Ni₂S₂ Ni@2c (site point group mmm) does not. This filter obtains 542 entries remaining. Third, the considered orbital multiplets are partially occupied within 1.0 eV around the Fermi level, leading to 71 material candidates as the “best of class”.

Reviewers' Comments:

Reviewer #1:

Remarks to the Author:

The authors have improved the discussion of the physics behind the enhancement of spin-orbit coupling effects by correlations and convincingly responded to the criticism.

In my opinion the importance of correlation enhanced SOC gaps is both stressed more convincingly and better explained than in the available literature. I agree with the other referee that the discussion is most transparent for the ferromagnetic materials, and perhaps could be improved somewhat still for the paramagnetic case -- but -- given that: 1) the ferromagnetic case is the case of interest to the manuscript, and 2) understanding well one limiting case is important, I feel the manuscript is robust enough from that point of view.

Most importantly, the demonstration of the effect on a set of materials relevant to topological insulators and classification of the problem from the symmetry point of view certainly is novel and interesting.

I recommend the paper for publication.

Reviewer #2:

Remarks to the Author:

I have read the answers to both referee reports. I think that the authors have done a quite nice job in answering all the questions. They have also done some additional calculations, and show more details in the supplementary materials. I appreciate that the authors did the effort to consider a general Kanamori Hamiltonian in the general case without full spin polarization. The results are convincing, and are presented nicely. I do not have more comments on this revised version and can recommend its publication. I think that there are quite a number of proposals in this manuscript for new topological materials. Whether they will prove useful in the lab and survive experimental verification, we will see. But this work has the potential to trigger quite a number of follow-up studies.